biophysics/statistical physics/computational physics

genotype–phenotype map, gene duplication, self-assembly, Polyomino

**Author for correspondence:**
V. Jouffrey
e-mail: vatj2@cam.ac.uk

# Gene duplication and subsequent diversification strongly affect phenotypic evolvability and robustness

V. Jouffrey[1,2], A. S. Leonard[1,2] and S. E. Ahnert[1,2]

[1]Cavendish Laboratory, University of Cambridge, JJ Thomson Avenue, Cambridge CB3 0HE, UK
[2]Sainsbury Laboratory, University of Cambridge, Bateman Street, Cambridge CB2 1LR, UK

 VJ, 0000-0002-6786-1484; ASL, 0000-0001-8425-5630;
SEA, 0000-0003-2613-0041

We study the effects of non-determinism and gene duplication on the structure of genotype–phenotype (GP) maps by introducing a non-deterministic version of the Polyomino self-assembly model. This model has previously been used in a variety of contexts to model the assembly and evolution of protein quaternary structure. Firstly, we show the limit of the current deterministic paradigm which leads to built-in anti-correlation between evolvability and robustness at the genotypic level. We develop a set of metrics to measure structural properties of GP maps in a non-deterministic setting and use them to evaluate the effects of gene duplication and subsequent diversification. Our generalized versions of evolvability and robustness exhibit positive correlation for a subset of genotypes. This positive correlation is only possible because non-deterministic phenotypes can contribute to both robustness and evolvability. Secondly, we show that duplication increases robustness and reduces evolvability initially, but that the subsequent diversification that duplication enables has a stronger, inverse effect, greatly increasing evolvability and reducing robustness relative to their original values.

# 1. Introduction

Over the past three decades, many computational models of genotype–phenotype (GP) maps and fitness landscapes have been put forward [1–3]. They offer a perspective on the study of biological evolution that is complementary to experimental work. GP maps are an abstract description of the way in which biological sequences (genotypes) map to different biological outcomes (phenotypes). The analysis of their structure reveals important information about the local properties of genetic

neighbourhoods. As a genotype accumulates mutations, it moves in genotype space, which is often defined as the network of possible sequences, with one-point mutations as the connections that link them.

In this work, we study the role that non-determinism and duplication play in abstract GP-maps of biological self-assembly.

Gene duplication has been shown to be pervasive among eukaryote organisms, suggesting a key role in evolutionary dynamics [4]. As opposed to mutations, which explore the local genetic neighbourhood, duplication fundamentally expands the entire genotype space and likely alters any original neighbourhood. An obvious consequence of duplication is the possibility to explore new features while conserving existing ones. As such, duplication can play a core role in evolution [5–7]. However, it remains unclear how the distribution of phenotypes in genotype space is affected by duplication.

With the advent of genomics and high-throughput techniques, experimental and computational biologists have accumulated phylogenetic evidence of large gene families that have originated from a single original ancestor. Multiple biological processes have been identified as enabling gene duplication, complicating the task of evolutionary biologists to model and assess its role and impact as a driver of evolution. Some gene families result in the formation of heteromeric protein complexes in which different protein subunits correspond to different members of the gene family [8,9]. The Polyomino model is well-suited to investigate how protein subunits form complexes as well as how the duplication of one of the genes may affect the protein complex self-assembly process [6].

To address this uncertainty, this work extends a Polyomino model of lattice self-assembly [10]. This model is a coarse-grained approximation of protein quaternary structure, and has been used to study the properties of protein complex assembly and the biological evolution of self-assembly more generally [11,12]. It allows tractable analysis of complete GP maps [1,13], which is not feasible in experimental settings beyond short sequences (greater than 10 nucleotides) [14] even with modern high-throughput technologies [15].

Gene duplication can be followed by mutations of the duplicated genes, which in turn means that two different versions of the gene may both interact with other genes or proteins in similar ways. In the context of self-assembly, and of the Polyomino model in particular, this may give rise to non-deterministic assemblies in which two similar proteins can interchangeably interact with another subunit. In order to study duplication, the Polyomino model has therefore been extended to incorporate non-determinism. Previous versions of the model regarded any non-deterministic or unbound assembly as a single deleterious phenotype [1,11]. In addition, new measurements of robustness and evolvability have been developed, based on sets of assembled structures rather than single structures (see Methods).

We find a subset of genotypes which have positively correlated robustness and evolvability. This is remarkable because these quantities are negatively correlated in the deterministic case, both in our analysis and in previous work [1,7,16]. Other studies have shown how this paradox can be resolved by focusing on population genetic considerations [17–19]. These results also highlight the fundamental role of non-determinism and the limitations of current GP-map models, which are typically deterministic. In the proposed model, non-determinism refers to the possibility that some genotypes may translate into more than one Polyomino, resulting in phenotypes with a variable number of features. Non-determinism is important in the context of duplication, and has been identified as playing an important role in evolution [20–22]. It is central to a more complete understanding of GP maps and evolutionary landscapes.

We also show that gene duplication tends to reduce evolvability for all phenotypes. The impact of duplication on robustness is however more contrasted. While a subset of phenotypes seem to benefit from duplication, the rest is negatively impacted. This reduction of evolvability and to some extent robustness seems to contradict the fact that there is ubiquitous evidence of gene duplication in eukaryotic evolution. In the final section of the paper, we offer a possible lead to solve this apparent contradiction. We show in two examples that the diversification that follows gene duplication can increase evolvability and reduce robustness. Thus gene duplication can yield phenotypes that are more robust (before diversification) *or* more evolvable (after diversification). These results remain to be confirmed by large-scale analysis in better-suited models.

# 2. Methods

## 2.1. The non-deterministic Polyomino model

The standard Polyomino self-assembly model takes a string of integers (each encoding the interface type of a particular face on a square tile) as a genotype, and assembles a Polyomino (set of connected lattice

sites) as a phenotype [1,13]. The GP map thus describes the mapping of all allowed integer sequences into Polyominoes.

This model is a coarse-grained representation of protein complex self-assembly, where proteins bind together to form larger structures. The coarse-graining procedure disregards the details of the secondary and tertiary structure of the protein as well as the molecular forces responsible for protein binding. Each three-dimensional interface is replaced by a single integer label identifying the type of interaction. Each specific label (except one neutral one) interacts with one other label. All interactions are attractive. As the internal structure of protein is lost, genetic mutation now means a mutation of the integer interaction labels, leading to the loss or gain of binding interactions between the subunits.

### 2.1.1. Genotypes and assembly graphs

The genotype is a string of $4T$ integers, encoding the interface types of each of the faces of $T$ square assembly tiles in a clockwise fashion. Interactions are purely attractive, and are defined in a pairwise manner. Interface type 0 is neutral and has no interacting partner, while type 1 binds to type 2, 3 to 4, etc. Tiles have cyclic symmetry in their notation, e.g. (1, 2, 0, 0) represents the same tile as (0, 1, 2, 0), (0, 0, 1, 2) and (2, 0, 0, 1).

Genotypes with multiple tiles also exhibit permutation symmetry because the order of tiles does not matter, e.g. the two-tile genotype {(1, 1, 1, 1), (2, 0, 0, 0)} and {(2, 0, 0, 0), (1, 1, 1, 1)} are the same set of tiles as far as the assembly process is concerned. These symmetries are an important contributor to the tractability of the Polyomino GP map, as discussed later on in this paper.

Interactions between interfaces can be represented using an assembly graph [23], which captures all the possible interactions between the tiles. This is a graph in which the faces of the tiles are the nodes, and interactions between interfaces are undirected edges. For the purposes of topological analysis, the faces of a given tile can also be thought of as being connected by a directed cycle of four edges, however, these directed edges are omitted in the figures of assembly graphs shown in this paper, as they are implied by the geometry of the tiles. The aforementioned symmetries within the tiles and genotype can be captured by identifying isomorphic graphs. These isomorphisms also capture a third symmetry, which arises because interaction pairs can be swapped. For example swapping 1 and 2 in a genotype yields the same assembly graph and assembly process, as does a swap of 1 and 3 together with a swap of 2 and 4.

### 2.1.2. Assembly and Polyominoes

The assembly process is stochastic and starts with a randomly selected tile as the seed. At every time step, an interacting face on the perimeter of the structure is chosen, and an interacting partner to that interface is chosen randomly from the assembly graph. The new tile binds irreversibly to the structure, and the process repeats with a new interacting interface chosen until only neutral faces remain on the perimeter of the structure. Interfaces for which the counterpart is not present in the genotype (e.g. genotype contains type 3 but not 4) are *de facto* neutral, and may also appear on the perimeter at the end of assembly.

Some assembly graphs will never reach the neutral perimeter state, growing endlessly through repeated patterns of growth. Such graphs are termed *unbound*, while their finite terminating counterparts are *bound*. As the assembly process is stochastic, it is possible for some graphs to produce multiple distinct Polyominoes, a direct manifestation of *non-determinism*. Only graphs which produce the same Polyomino every time are called *deterministic*. Some examples of genotypes and their corresponding assembly graphs, and the shapes they assemble into are shown in figure 1. In the following, we will distinguish assembly graph phenotypes $\{P_a\}$ and shape phenotypes $\{P_s\}$. The assembly graph that given a genotype maps to is an element of the former, and the assembled shape (or set of shapes) is an element of the latter.

There are infinitely many copies of each tile available in the assembly process. Absolute rotations or translations of the Polyomino are not meaningful, and we choose to not distinguish between chiral counterparts, i.e. reflections through the two-dimensional plane. Polyominoes obeying these symmetries are called *free* Polyominoes.

### 2.1.3. Phenotypes as sets of Polyominoes

The neighbourhood of duplicated genes may contain a significant number of genotypes with non-deterministic shape phenotypes, often due to mutations that affect one of the copies of the duplicated

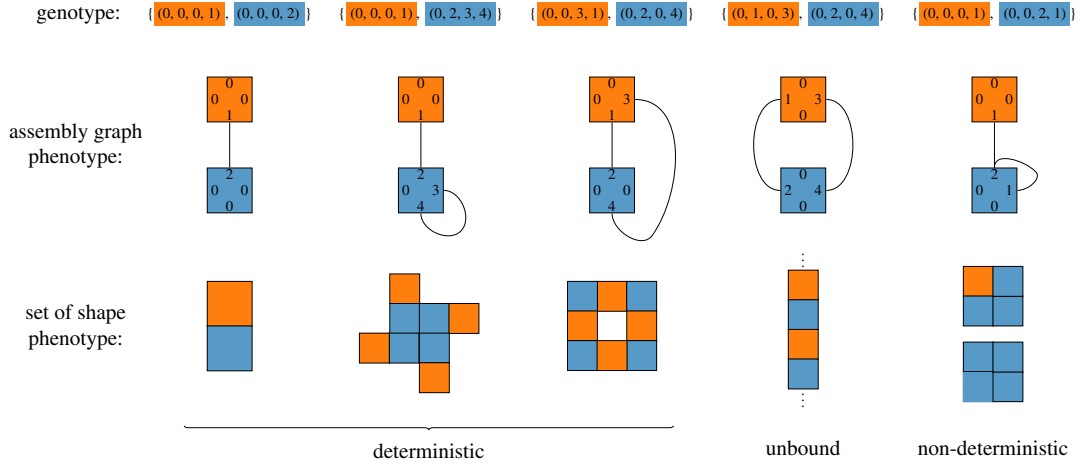

**Figure 1.** Examples of genotypes, assembly graphs and phenotypes. The first three examples are deterministic, the fourth example shows an unbound shape, while the final one is non-deterministic, as it can give rise to two different Polyominoes. The interacting label pairs are (1, 2) and (3, 4), meaning edges labelled by a 1, respectively 3, attach to edges labelled by a 2, respectively 4.

gene. The mutated genotype could either produce the original shape using the unmutated gene or a new shape using the new version of the gene.

Modern phylogenetic techniques have identified gene families which consist of structurally (and usually functionally) related genes with a common evolutionary origin. Despite the gene sequence similarity, the corresponding proteins form different complexes, e.g. heme proteins from the globin family [24]. The deterministic version of the Polyomino model allows only one Polyomino per genotype. The heme gene family example as well as numerous other ones highlight however that we should also consider the diversification of protein complexes that can be formed, meaning the emergence of coexisting variants of a given complex through duplication and subsequent specialization. In other words, it shows how the restriction of a phenotype to a single Polyomino is ill-suited to the study of gene duplication as the restriction to deterministic phenotypes inhibits the diversification of protein complexes. This process naturally shows how the restriction of a phenotype to a single Polyomino is ill-suited to the study of gene duplication. It is therefore natural to define the shape phenotype as a set of reliably produced Polyominoes. This definition has the advantage of extending the phenotype to non-deterministic cases in a way that incorporates deterministic Polyominoes as a subset.

Reliability can be defined using the requirement that a Polyomino is assembled more than a certain fraction of times, e.g. 10%, from a given genotype in order to contribute to the phenotype. In cases where a Polyomino is produced less often than this, it is combined alongside any other infrequent Polyominoes into a single *unstable* subset. In all the analysis that follows the required fraction is set to 25%. The results presented vary slightly depending on the fraction in the 10 to 25% range, and the qualitative observations hold in general. Any assemblies that are unbound are grouped into a single 'unbound' subset, regardless of the assembled structures. This includes cases where the assembly process is non-deterministic and leads to both bound and unbound shapes. The bound shapes produced are discarded and the phenotype is solely assigned an unbound label.

## 2.2. Genetic neighbourhood properties

In the deterministic case, evolvability and robustness are negatively correlated at the genotype level [1,7]. This is because genotypic robustness in this context is defined as the fraction of single-point mutations of the genotype that leave the phenotype unchanged, and genotypic evolvability is defined as the number of different phenotypes in the single-point mutation neighbourhood. Since a genotype cannot simultaneously be surrounded by a large number of neutral genotypes and a large number of genotypes of different phenotypes, the two quantities cannot both be large within the range of allowed values.

This negative correlation is observed across several GP maps, such as that of RNA secondary structure [25] and toyLIFE [26], as well as emperically derived maps [27]. This correlation is also found in the Polyomino model [1], and is likely to be a general property of GP maps [28–30].

The extension of shape phenotypes from individual Polyominoes to sets of Polyominoes in order to capture non-deterministic effects requires new definitions of evolvability and robustness. Each genotype in the genotype space has $4T(C-1)$ single-point mutation neighbours for a $T$-tile Polyomino system with $C$ interface types. The metrics introduced below aim to provide continuity with earlier work on the Polyomino model where possible, but will also diverge from this prior work due to the new definition of phenotypes as sets of Polyominoes, and also due to the focus of this study on the impact of duplication on local genetic neighbourhoods.

Genotypic set robustness is defined as the proportion of neighbours which can reliably produce at least one Polyomino present in the original shape phenotype set $\mathcal{P}_{\text{ref}}$, defined as

$$r_g = \frac{\sum_n [\mathcal{P}_n \cap \mathcal{P}_{\text{ref}} \neq \varnothing]}{4T(C-1)},$$

where [.] are Iverson brackets taking values of 1 or 0 if the internal condition is true or false, respectively, corresponding to the fraction of neighbours with a non-null phenotype intersection.

Genotypic set evolvability is defined as the proportion of neighbours which reliably produces at least one new shape that is not in the reference set, i.e. the set difference, defined as

$$e_g = \frac{\sum_n [\mathcal{P}_n \setminus \mathcal{P}_{\text{ref}} \neq \varnothing]}{4T(C-1)},$$

set evolvability measures the number of new shapes added to the original phenotype.

In the deterministic case, each neighbour can only contribute to set robustness or set evolvability leading to competition between the two quantities. However, non-determinism allows phenotypes that are a set of several Polyominoes, leading to the possibility that a neighbour contributes to both quantities. This occurs if a neighbour conserves the original Polyomino(es) as well as adding new ones. This overlap, which is impossible in a deterministic context, encapsulates the added evolutionary potential that non-determinism offers. The set robust-evolvability of a genotype ($s_g$) is defined as the fraction of neighbours contributing both to robustness and evolvability.

$$s_g = \frac{\sum_n [[\mathcal{P}_n \cap \mathcal{P}_{\text{ref}} \neq \ ] \wedge [\mathcal{P}_n \setminus \mathcal{P}_{\text{ref}} \neq \varnothing]]}{4T(C-1)},$$

where $\wedge$ denotes a logical AND operation. However, not all shapes which can be built stochastically from a given genotype may be built with the same frequency. To capture this imbalance, we introduce a minimum threshold frequency with which a shape must be built in order to consider it as being produced reliably. It is set to 25% in the results presented throughout this paper. Any shape built less often is considered rare, and a generic 'rare' placeholder is added to the phenotype instead of the shape. We define the fraction of local neighbour shape phenotypes that include a non-deterministic placeholder as the *genetic instability* $g_g$:

$$g_g = \frac{\sum_n [U_{\text{rare}} \in \mathcal{P}_n]}{4T(C-1)},$$

where $U_{\text{rare}}$ is the 'rare' placeholder. Neighbours that assemble into any unbound structure are classed as unbound, as this behaviour is often deleterious in biological contexts, such as uncontrolled protein aggregation. The unbound fraction in the local neighbourhood of a genotype is defined as

$$u_g = \frac{\sum_n [U_\infty \in \mathcal{P}_n]}{4T(C-1)},$$

where $U_{\text{inf}}$ represents any unbound structure. A schematic neighbourhood that illustrates these classes is shown in figure 2.

We can define phenotypic equivalents of $r_g$, $e_g$, $s_g$ and $u_g$ for the assembly graph phenotypes $\{P_a\}$ by calculating the mean of each of these quantities over all genotypes $g$ that map to a specific assembly graph phenotype $p \in \{P_a\}$. For example

$$r_p = \frac{1}{|S_p|} \sum_{g \in S_p} r_g,$$

where $S_p$ is the set of genotypes that map to assembly graph phenotype $p$, and $|S_p|$ is the size of $S_p$. These averaged quantities will be helpful in describing the statistical properties of the GP-map. There is a close (and in many cases one-to-one) correspondence between assembly graph phenotypes $\{P_a\}$ and set of shape phenotypes $\{P_s\}$. The latter make it possible to define non-deterministic versions of

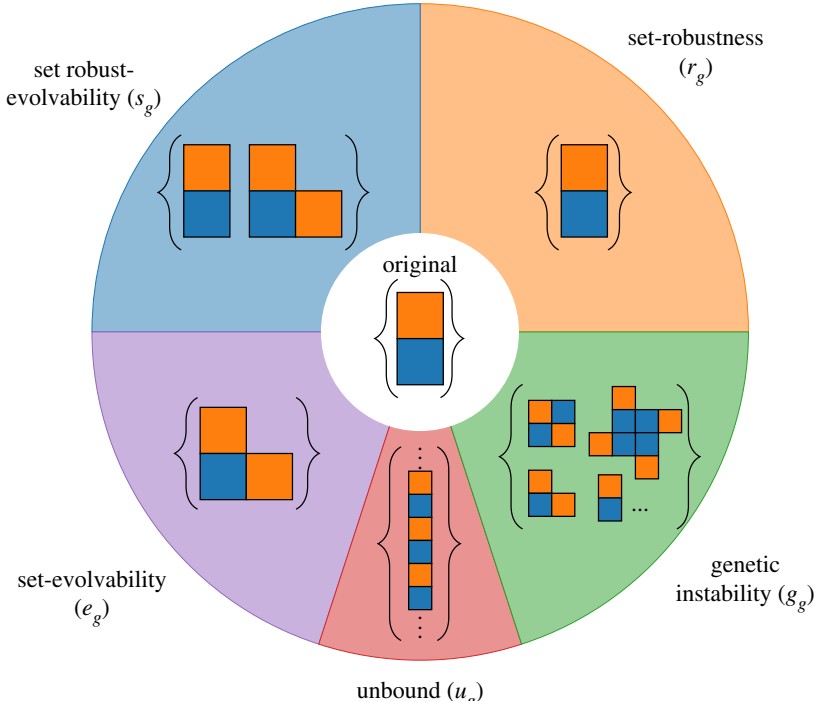

**Figure 2.** Schematic description of the local neighbourhood of a heterodimer. Mutational neighbours capable of forming a heterodimer contribute to set robustness, while neighbours that form other shapes contribute to set evolvability. Due to the presence of non-determinism, neighbours may contribute to both metrics if they form a heterodimer and another shape reliably. The corresponding neighbours are double counted for both set-robustness and set-evolvability. This overlap is denoted set-robustness and evolvability and is expected to be key in the study of the impact of gene duplication. Neighbours which produce rare shapes contribute to genetic instability while unbound shapes are in a separate category.

robustness and evolvability at a genotypic level, and the former make it much more tractable to calculate phenotypic averages. There is a limited number of cases in which several assembly graph phenotypes correspond to a single set of shape phenotype.

Full details on this approach can be found in supporting information, appendix.

# 3. Results and discussion

GP maps are a space in which evolutionary trajectories can be observed. They need to be combined with appropriate fitness functions and population dynamical models to fully describe evolutionary processes. However, the GP map does constrain the possibilities and probabilities of these trajectories. In particular, determining the robustness and evolvability of local genetic neighbourhoods informs evolutionary models by providing insights into the probabilistic effects of evolutionary processes such as mutation or gene duplication. Indeed, one may calculate the average robustness and evolvability of all the genotypes with given phenotypes to provide the probabilities to evolve or persist upon mutation in a population dynamic model for example.

The first part provides a statistical analysis of the distribution of accessible phenotypes upon a single mutation of a genotype. This step is necessary in order to characterize the local properties of the full GP map, before focusing our analysis on gene duplication. Indeed, the addition of a gene to a genotype alters its point-mutation genetic neighbourhood. It is therefore key to understand the properties prior to duplication in order to understand the consequences of this altered local neighbourhood on the accessible phenotype distribution. These results directly inform future models of evolutionary dynamics by providing insights upon how evolvability and robustness are affected by duplication events. Indeed, we show how mutations before and after duplication display different probabilities of changing the phenotype. These probabilities and their relationship are key parameters in population genetic models and evolutionary dynamic studies.

This relationship between robustness and evolvability has been the focus of numerous experimental and theoretical investigations. Particular attention has been devoted to linking the local properties of GP

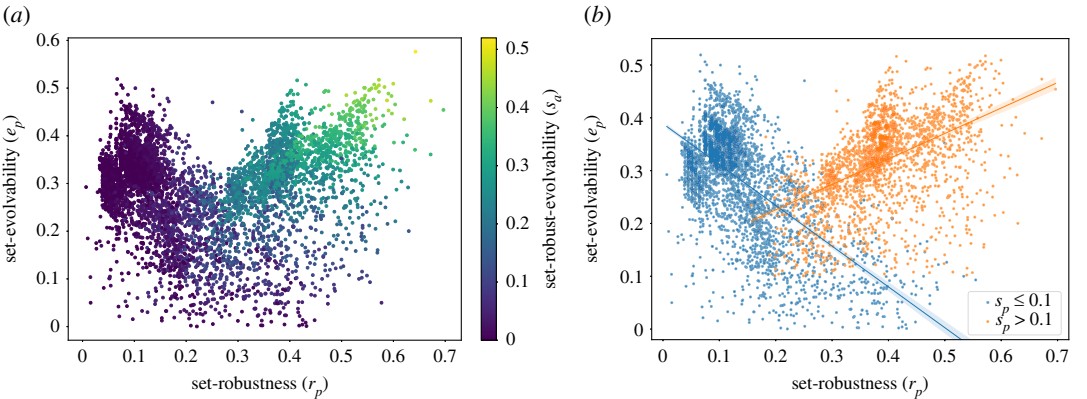

**Figure 3.** Scatter plot of the phenotype-averaged set robustness versus phenotype-averaged set evolvability for the four-tile Polyomino GP map. On the left-hand side, the colour scale indicates $s_p$, which is the phenotype-averaged fraction of neighbours that contribute to both set evolvability $e_p$ and set robustness $r_p$. On the right-hand side, the phenotypes are subsequently divided into two sets based on a $s_p$-threshold chosen to lie in between the two maxima of the $s_p$ distribution (see §5). It highlights a qualitatively distinct correlation pattern between $r_p$ and $e_p$ for the low- and high-$s_p$ groups.

maps at the genotype level to robustness and evolvability at the population scale [31,32]. While robustness and evolvability are modelled as mutually exclusive possibilities on a genotypic level, studies have shown that a positive correlation between robustness and evolvability emerges at the phenotypic scale.

The results presented here depart from previous work [31,32] as they rely on fundamentally different assumptions. The addition of non-determinism to the model enables an overlap between robustness (conserving a phenotypic trait) and evolvability (discovering a different trait) at the genetic level. Instead of thinking of mutation or duplication as simply changing the phenotype, the focus here is on the addition or loss of phenotypic features. This also means we have to define robustness and evolvability accordingly.

While this non-deterministic model has wider implications that would be interesting to explore, the emphasis here is on the effect of gene duplication on the structural properties of the GP map. The statistical analysis aims to provide a large-scale perspective, but does not permit conclusions at the population level. Rather, it provides a starting point for the definition of local GP map properties that work in the context of gene duplication as well as single point mutations (figure 3).

## 3.1. Evolvability and robustness in non-deterministic genotype–phenotype maps

In this section, we aim to characterize the impact of including non-determinism in the Polyomino self-assembly model. While non-deterministic genotypes were always part of the Polyomino GP-map, previous models rejected the contribution of genotypes that assembled into more than one Polyomino. The results presented in figure 3 show how the relaxation of this constraint changes the negative correlation between set robustness and set evolvability that is inevitable in deterministic GP-maps. We illustrate that non-determinism gives rise to a subset of phenotypes that have both high set robustness and high set evolvability, and that a positive correlation between the two metrics emerges for this subset.

Figure 4$b$ shows the set evolvability $e_p$ and set robustness $r_p$ of all four-tile phenotypes with up to four interface types, with the value of $s_p$ indicated by the colour of the points. As $s_p$ indicates the fraction of mutational neighbours that contribute to both evolvability and robustness we can see that the phenotypes fall into two classes: firstly, low or zero $s_p$ phenotypes (shown blue in the right panel of figure 4$b$) for which robustness $r_p$ and evolvability $e_p$ are negatively correlated, and secondly, high $s_p$ phenotypes (shown orange in the right panel of figure 4$b$) for which $r_p$ and $e_p$ are positively correlated.

In previous work, which only considered deterministic Polyomino self-assembly [1,10,11,13], non-deterministic and unbound assembly was mapped onto a single phenotype (and usually regarded as deleterious). In the framework proposed in this paper, which allows non-determinism, we characterize unbound assembly by labelling any genotype that is capable of building an unbound structure as 'unbound', even if the same genotype builds bound structures most of the time. The unbound fraction $u_g$ measures the fraction of unbound genotypes in the single-point mutation neighbourhood of a

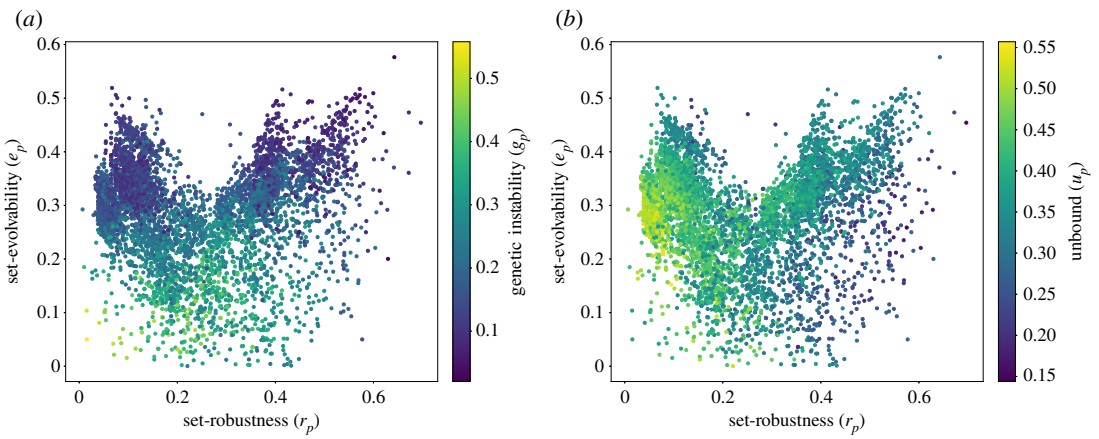

**Figure 4.** Scatter plot of the set robustness versus set evolvability for the four genes GP-maps. On the left-hand side, the colour scale indicates the percentage of neighbours contributing to genetic instabilities. On the right-hand side, the colour scale indicates the percentage of neighbours producing unbound shapes. Note how high set robustness and set evolvability genotypes may still have a relatively high fraction of unbound neighbours. This is a direct consequence of non-determinism, with other neighbours capable of contributing to both set robustness and set evolvability. The regression data are provided in supporting information, appendix.

given genotype. The phenotypic measurement $u_p$ averages this over all genotypes that belong to that phenotype.

Non-zero $s_p$, meaning the existence of mutational neighbours that contribute to both robustness and evolvability, is only possible if non-deterministic self-assembly phenotypes are allowed. Non-determinism thus enables phenotypes to be both robust and evolvable. In addition to this advantageous property however, non-determinism can also mean that a given set of shapes is not produced reliably. The quantity $g_p$ measures this instability, and its distribution across the phenotypes is shown in figure 4. Phenotypes with low $e_p$ (and to a lesser degree, low $r_p$) exhibit high values of $s_p$.

## 3.2. Impact of gene duplication on local phenotype distribution

In the previous section, we presented metric results for each phenotype in the GP map providing a statistical characterization of the full GP map. In this section, we focus on the impact of gene duplication and the statistical characteristics of a subset of the GP map containing only genomes which have a duplicated gene. As such, GP map level properties may not match properties observed for full GP maps. The analysis relies on duplicating a single gene on all the assembly graph of the 4-gene GP map and performing a similar statistical analysis. The results are then compared to the 4-gene GP map to determine the effect of duplication.

As can be seen in figure 5, the overall impact of duplication on the set evolvability metric is negative with a global average loss of 0.076. However, there are also genotypes which receive an increase in evolvability from duplication.

The impact of duplication on set robustness is more nuanced. Figure 5 shows that the impact on set robustness depends on the fraction of neighbours that contribute to both set robustness and set evolvability of the original genotype. Indeed, genotypes with high $s_g$ seem to experience a decrease in $e_g$, and those with low $s_g$ an increase.

The results presented in figure 5 suggests a link between the impact of duplication on robustness and—through $s_g$, which relies on non-deterministic phenotypes—the amount of non-determinism present in the local neighbourhood of a genotype. It underscores the importance of including non-deterministic effects when studying gene duplication at the level of protein self-assembly.

## 3.3. Duplication and interface specialization

By considering particular example phenotypes, this section focuses on the notion of interface specialization after a duplication event. The emphasis is on the statistical differences in the genetic neighbourhood of genotype categories: original, duplicated and interface specialized. The analysis is

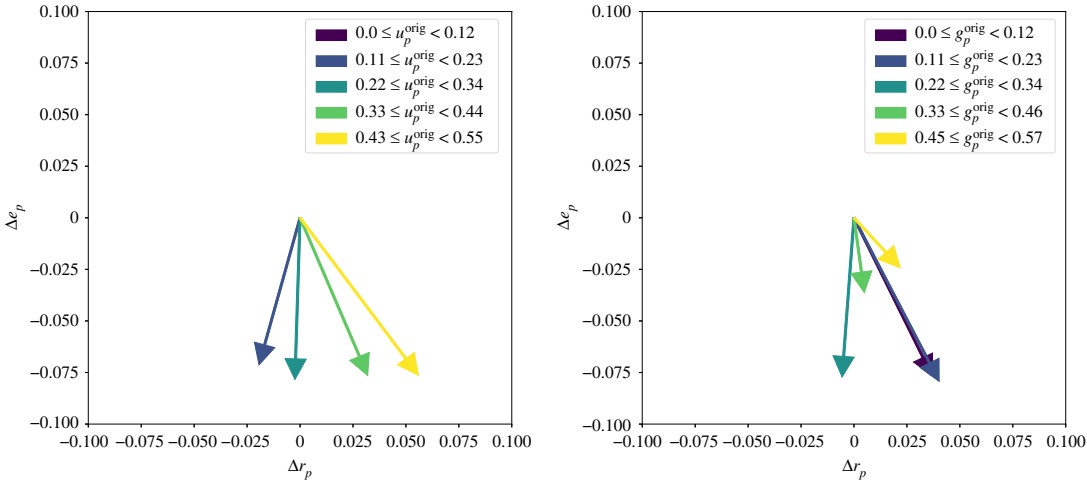

**Figure 5.** Impact of gene duplication on the phenotype of the 4 genes GP-map. Arrows indicate the average change in $r_p$, $e_p$, $s_p$ or $g_p$. The duplication data contains over 16 000 points and have therefore been binned and averaged to improve readability. The data points are binned depending on the value of a given metric in the original 4 genes GP-map.

strictly static and no evolutionary dynamic is involved. Instead, genotypes are assigned a category based on their assembly graph (figure 6).

Interface specialization is a direct consequence of the independent neutral drift of the two copies of a gene. Consider the following example. Pre-duplication, a protein has two domains binding to two copies of a second protein. Both interfaces are constrained to have the same binding sites to fit the same second protein. Post-duplication of the gene encoding the second protein, such a constraint is lifted. Indeed, the two copies of the gene drift independently of each other potentially leading to a specialization of each interface.

In the Polyomino model, the corresponding phenotype is a trimer and in its simplest form can be built using only two elementary bricks and a single interacting label pair. In the two-tile GP map, there is a single assembly graph leading to such a shape. In the three-tile GP map, there are three different assembly graphs which can lead to this shape. Two of them correspond to the previous genotype, albeit with an extra copy of one of the genes. The third assembly graph corresponds to a genotype which involves one more interacting label pair. Using an extra interaction pair represents the extra degree of freedom which can be granted by independent neutral drift of the two copies. The assembly graph for all three genotype groups is illustrated in figure 6.

In both studied cases, duplication reduces set evolvability and increases set robustness (see figure 7). On the same figure, one may observe that subsequent specialization however has the opposite effect, increasing set evolvability greatly while reducing set robustness and genetic instability.

The specialization of each interface in the diversified tile set results in a steep drop in set robustness, which could be partly explained by the loss of redundancy in the specialized genotype. One must also consider a potential 'interference' effect due to the third tile introduced by the duplication-specialization chain. Indeed mutations previously contributing to robustness are now leading to a different shape due to the presence of the third tile. As an example, consider mutating a neutral 0 label to a 4 in the original genotype versus the specialized version. The presence of the third tile leads to a tetramer in the specialized version whereas it does not impact the original one. This process also decreases the number of neighbours forming rare shapes, pointing to a more stable building assembly process. This is in line with the 'duplication-degeneration-complementation' model [33], where in the post-subfunctionalization genotype, genes are more susceptible to negative mutations as both the now unique tiles are required for successful assembly.

Additionally, after subfunctionalization, the average genotype is now highly evolvable, and is able to add new features at a much higher rate [34], recovering observations about the combined role of subfunctionalization and then neofunctionalization [35,36]. The higher set evolvability of the diversified tile set is likely due to the fact that the larger variety of tiles allows more subsets of tiles to become new phenotypes.

The results presented in figure 8 suggest an interesting connection between gene duplication and non-determinism. Indeed, post-duplication data highlight an increased tendency to form rare shapes

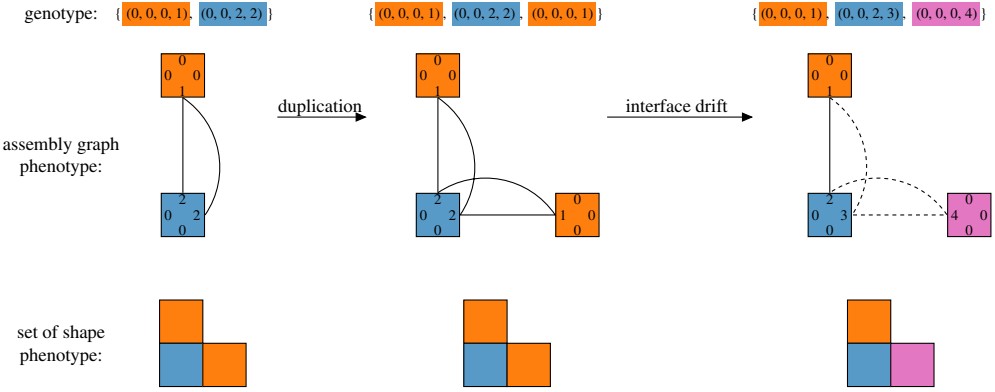

**Figure 6.** Assembly graph representation of a possible evolutionary sequence involving interface specialization of an heterotrimer. The parent genotype has only two genes, preventing the two binding sites from evolving independently to conserve the trimer structure. After duplication, both interfaces may become independent and their sequences slowly drift away from each other. Studying the dynamical aspect of these phenomena in the integer Polyomino model is problematic as it does not allow for slow coevolution of binding sites, instead the focus is on static analysis of neighbourhood properties for the three categories of assembly graph: original, duplicated and specialized.

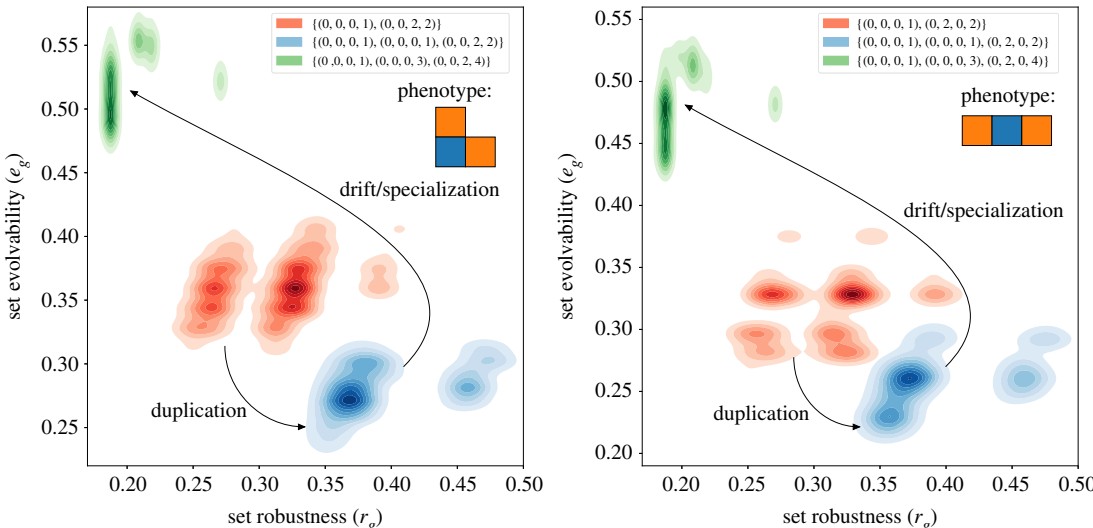

**Figure 7.** Kernel density estimation plots of robustness versus evolvability for the three genome groups of each of the L-shape and I-shape trimers: original, duplicated and specialized. The duplication step improves the robustness at the cost of lowering evolvability as discussed in the main text. The second step, corresponding to independent drift of the interface, causes a massive evolvability gain. This comes at the cost of losing the original robustness boost provided by the gene redundancy. While the precise metric values differ, the pattern is remarkably similar between both trimers.

in the local neighbourhood of the genotypes that possess a duplicated gene. This is contrasted by post-specialization data which indicates a reduced presence of genotypes forming rare shapes in the neighbourhood of specialized genotypes. The first observation is aligned with the expectation that duplication enables the accumulation of features at the phenotype level despite the stability issues, while the second observation would suggest that the specialization step provides stability of the self-assembly process at the expense of the robustness bonus provided by redundancy. This contrast points towards an interesting and intricate link between the duplication-specialization process and non-determinism. While only indicative, these results are in agreement with the larger studies presented in the first part of the discussion where non-determinism appeared to be distributed unevenly across the GP map.

We emphasize here that the model and methodology presented in this paper does not allow for a systematic study of interface specialization. As exemplified in this section, the minimum number of mutations for a genotype to show interface specialization is two. As such, the single-mutation neighbourhood analysis performed after duplication cannot capture such neighbours. New methods

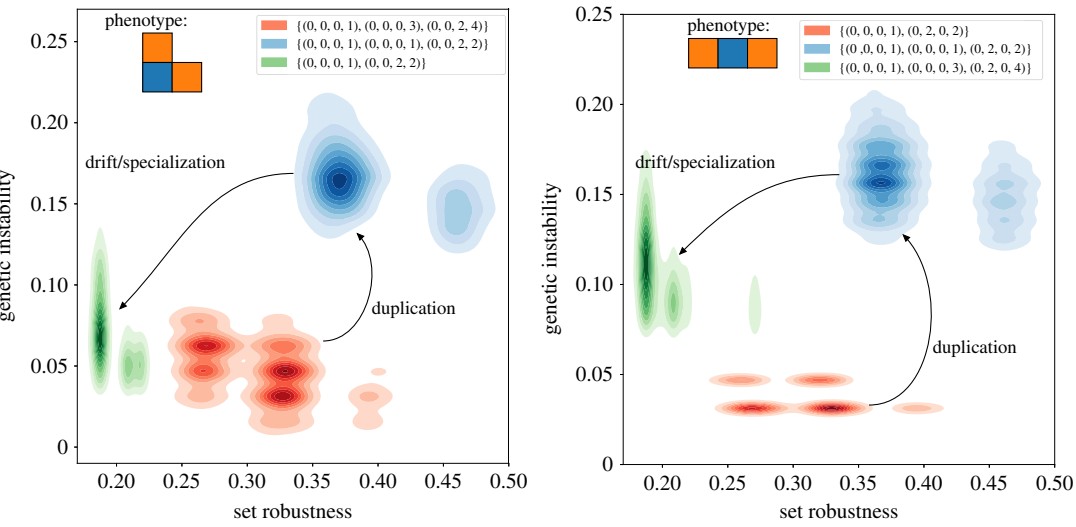

**Figure 8.** Kernel density estimation plots of robustness versus genetic instability for the three genome groups of each of the L-shape and l-shape trimers: original, duplicated and specialized. The duplication step massively increases the neighbourhood genetic instability. This emphasizes the prevalence of non-determinism in genomes that include a duplicate gene. The second step, corresponding to independent drift of the interface, strongly decreases the genetic instability to a level similar to the pre-duplication value. This highlights the fact that the non-determinism increase seems not due to a genotype size increase but to duplication. Again, the pattern is remarkably similar between both trimers.

and additional studies are therefore needed to shed light on the complementarity of these two mechanisms as well as their impact on GP-maps.

## 4. Conclusion

The non-deterministic Polyomino model allows for the study of gene duplication at the level of a full GP map. Using efficient sampling methods, one may infer the statistical properties of the local neighbourhood of genotypes for each phenotype. Characterizing the local properties of a GP-map is essential to understand the statistical impact of different evolutionary processes such as gene duplication. Such characterization is made possible by introducing a generalized metric suitable for non-deterministic models.

The results demonstrate the heterogeneity of non-deterministic genotypes' distribution leading to an uneven impact across the GP-map. Additionally, different non-deterministic genotypes behave distinctly leading to either positive or negative impacts on the set robustness and set evolvability. In particular, neighbours contributing to both set robustness and set evolvability offer the possibility for genotypes to score highly on both metrics. The latter are relevant to better understand evolutionary dynamics, suggesting non-determinism at the GP-map level might be an essential ingredient in evolutionary models.

Our findings show gene duplication as having an overall negative impact on set evolvability; the qualitative impact on set robustness seems to depend on the fraction of genotypes in the original neighbourhood contributing to both metrics. These results contrast with empirical evidence which reveals the ubiquity of gene duplication across eukaryote organisms. However, the results also reveal important disparities between genotypes, with some benefiting from both increased set robustness and set evolvability while others may have one or both metrics impacted negatively. Further studies are necessary to determine if this disparity may result in the prevalence of phenotypes which benefit from gene duplication at the expense of others.

The last section offers a complementary path to conciliate reality and theoretical results. Indeed, the analysis focuses on the possibility of interfaces specializing after duplication leading to a local neighbourhood with different properties. While the scope of this analysis is limited, the trimer example shows in particular that the impact of duplication and subsequent specialization seems somewhat opposed, suggesting a degree of complementarity. Further studies are necessary to study such complementarity with a more systematic approach.

# 5. Supporting information

## 5.1. GP map analysis through statistical sampling: methods

The purpose of this section is to introduce the various techniques which have been developed to explore efficiently the Polyomino GP map. The first method introduces a compression method which reduces redundancy in genotype space while preserving all phenotypes. The second part introduces a stochastic sampling method to compute metric distributions for each phenotype in the GP map.

## 5.2. Genotype space compression

This method aims to exploit the symmetries of the building assembly process to reduce the redundancies in genotype space. This compression of the genotype space is complete in the sense that all phenotypes present in the original GP map are represented by at least one genotype in the reduced set. Indeed, efficient use of symmetries can help manage the exponential growth of the genotype space in the Polyomino model as the number of gene or colour increases.

### 5.2.1. Genotype equivalence class in the pseudo-deterministic Polyomino model

Each genotype can be mapped on an assembly graph. The nodes of the latter are squares and the edges of the graph connect different square edges. Two identical assembly graph will build the same set of shape. Two genotypes will lead to the same phenotype if their assembly graph are identical. Note that it is not necessary for the graph to be identical to obtain the same set of shapes. Unfortunately, graph isomorphism is a complex operation and is not computationally tractable for spaces which include more than a million genotypes. However, it is possible to identify genotype operations which leave the assembly graph unchanged. These operations can be used to compress the genotype space before running a graph isomorphism algorithm.

The first operation is the invariance of a gene under cyclic permutation. Indeed, each gene is constituted of four number which label the edges of the square nodes in the assembly graph. Cyclic permutation amount to a $\pi/2$-rotation of the square. This rotation does not affect the graph, only its representation. Building the set of unique sequences under cyclic permutations from a given dictionary is well-known in combinatorics problem. This set is called the set of necklaces and can be build in polynomial time. A variant of this problem deals with adding mirror symmetry to the allowed transformations. This allows for a compression of the gene space and reduce significantly the number of genotypes to analyse.

Additional operations can be performed at the level of the genotype. The assembly graph is for example obviously invariant under gene permutations. Other operations involve genotype-wide label exchange. Those are due to the undirected nature of the edges in the assembly graphs. First, global genotype transformation which reverse all the labels for a given pair of interacting colours (e.g. all 1 become 2 and 2 become 1). Similar to this operation is an exchange of interacting pair (all 1,2 become, respectively, 3,4 and all 3,4 become, respectively, 1,2).

In the assembly graph, there are two possibilities for the edges of a square to be disconnected from any other edge. Either it is labelled with a neutral colour or it is labelled with an interacting colour but this colour does not appear in the rest of the genotype. So all unpaired interacting colours can be changed to a neutral colour or vice-versa.

This last operation is in some sense fundamentally different from the previous ones. None of the previous operation affects the metrics that will be used to analyse the GP map since the set of single mutation neighbours will include exactly the same phenotypes with the same frequency for the original or the transformed genotype. While the last operation also preserves the assembly graph, the local neighbourhood is *not* preserved. The single mutation neighbours may have different phenotypes with different frequencies.

## 5.3. Generate the set of minimal genootypes

The aim of this section is to use the various operations described previously to generate a compressed genotype space. The goal is to obtain a reduce genotype space which includes only a single or a few genotypes for each phenotype.

The first step is to build the allowed necklaces based on the number of interacting and neutral colours. The set of necklaces will constitute a gene pool from which the genotype can be built step by step. The gene pool will be further filtered depending on which genes are already included in the genotype prefix. The filtering will be essentially based on the relabelling transformations.

There are two main constraints imposed on the gene pool which will be referred to as growing constraints. These constraints will be imposed starting from the left. First, the first member of the pair appearing in the genotype must always be the odd number. Second, a pair can only appear in the genotype providing at least one member of all the lower pair appeared already. Both constraints can be implemented by keeping track of the maximum label already encountered in the genotype prefix and filtering the gene pool accordingly.

Finally, all genotypes which have at least one unpaired interacting colours are discarded. Indeed, unpaired interacting colours play the same role as the neutral colour in the assembly graph. The aim of the minimal set being to compress the genotype space, such redundancies need to be eliminated. This additional constrain is an important bias that will need to be properly addressed when using the minimal set to analyse local neighbourhoods and infer static GP map properties.

### 5.3.1. Isomorphic assembly graph

The number of genotype in the minimal set should scale roughly as the number of different phenotype which also scales exponentially. For small enough size, it is computationally possible to run an isomorphic check on the assembly graph associated to each genotype. One obtains an isomorphic set, which contains a single genotype for each unique possible assembly graph. The results in the following sections will make extensive use of the isomorphic set to explore the GP map properties.

Note that different graph may map on the same set of shape, therefore sampling the assembly graph is not strictly equivalent to phenotype sampling for set of shape. However, one could define the phenotype at the assembly graph level. Indeed, many genotypes map on a single assembly graph and an assembly graph maps on a unique set of shape (as opposed to a unique shape). There are cases where several assembly graphs map on the same set of shape, as there can be several assembly pathway leading to a single polyomino for example.

This section provides a method to build a numerical mapping between a reduced genotype space and the assembly graph space at the cost of essentially throwing away information about the genotype space structure. While mutations provided a natural measure to the original space, there is no corresponding structure in the assembly graph space. However, the mapping captures the diversity of phenotypes which exist in the GP map and is therefore useful as a reference genotype-phenotype database. For each phenotype, one may find its corresponding representant. The latter can help generate all other genotypes in its equivalence class using the transformation described previously. The generated genotypes will form the same phenotype but may have different local neighbourhoods.

The second part of this section will focus on using this database to sample the distribution of phenotypes on the hypercube structure induced by single mutation in genotype space.

## 5.4. Metric distribution sampling

The compression of genotype space was the first step in implementing a statistical phenotype sampling of the GP map. The compression was designed to group all genotypes which have the same assembly graph in the same equivalence class. However, no care has been taken has to distinguish between genotypes based on their single mutation neighbourhood. While the relabelling transformations discussed above accounts for a true redundancy in the Polyomino GP map model, other transformations do not. The assembly graph equivalence class group together genotypes which will have different neighbourhoods and therefore perform differently on the various metrics introduced in the main text. In order to explore the GP map properties, one must devise a sampling method which explores the distribution of metrics within an assembly graph equivalence class.

### 5.4.1. Uniform sampling via neutral label mutation

The aim of this section is to introduced a sampling method which allows the numerical evaluation of local and global GP map metrics.

The database constructed in the previous section will serve as a starting point for the stochastic sampling of the Polyomino GP map. Phenotype metrics are built out of the local metric computed for each of their

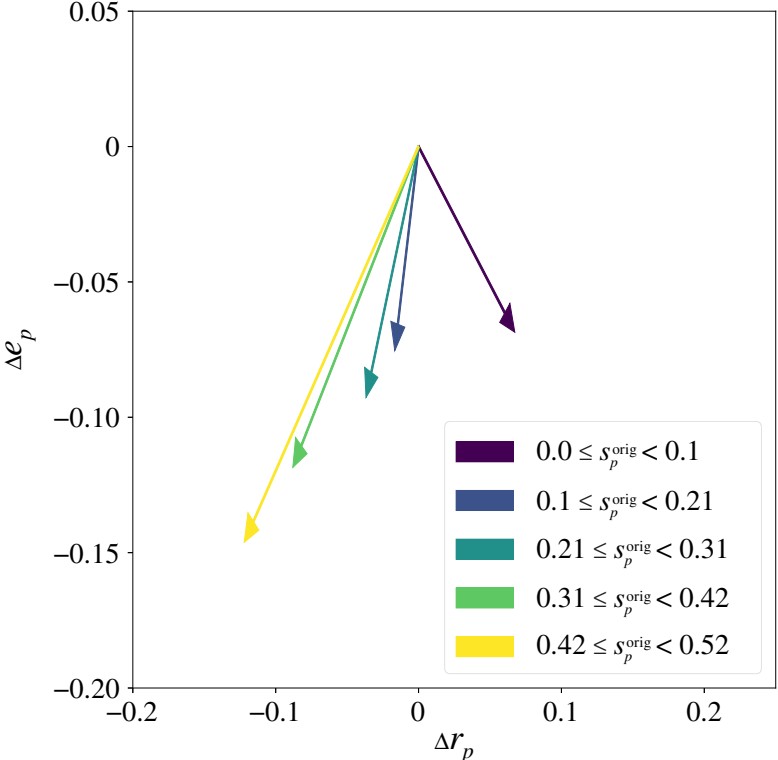

**Figure 9.** Impact of gene duplication on the phenotype of the 4 genes GP-map. Arrows indicate the average change in set robustness $\Delta r_p = r_p^{\text{dup}} - r_p^{\text{orig}}$ and set evolvability $\Delta e_p = e_p^{\text{dup}} - e_p^{\text{orig}}$. The duplication data contains over 16 000 points and have therefore been binned and averaged to improve readability. The data points are binned depending on the value of a given metric in the original 4 genes GP-map. It highlights clearly how original genotypes with high-$s_p$ are negatively impacted by gene duplication while low-$s_p$ genotypes show an averaged gain of set-robustness albeit with a lower set-evolvability.

corresponding genotypes. For each set of shapes, one may use the corresponding genotype in the isomorphic database to construct its equivalence class. Uniform sampling of the latter reveals the underlying metric distribution. One may also compute the phenotype diversity by progressively building the set of unique shapes encountered when analysing each genotype's neighbourhood.

In order to save computational time, one can sample the equivalence class without including the relabelling transformations. Indeed, the latter have no effects on the local metrics. Only mutation of the neutral labels will affect the neighbourhood of a genotype. To preserve the AG, one may only mutate neutral labels into unpaired interacting colours. Uniform sampling of the reduced equivalence class[1] can be achieved using the following method. First, one must build a set containing a single member of each interacting pair which is not present in the isomorphic genotype. Second, one can mutate each of the neutral label using random elements of the set. One must analyse the neighbourhood of each genotype which has been generated. Histograms can then be used to visualize the distribution of values for each equivalence class.

### 5.4.2. Weighted average of non-isomorphic graphs

Some phenotypes correspond to several non-isomorphic AG. In order to obtain the correct averages for phenotype level metrics, one must assess the relative weight of each assembly graph depending on the corresponding proportion of representing genotypes. As the number of colours and number of gene increase, more and more of the simplest phenotypes will correspond to several non-isomorphic AG. In general, simpler graph have less constraints and therefore are much more common due to the high redundancy of the genotype space via relabelling transformations. Stochastic sampling methods can take advantage of this simplification for normalized quantities. However, metrics such as evolvability

---

[1]Meaning genotypes generated via relabelling transforms are excluded.

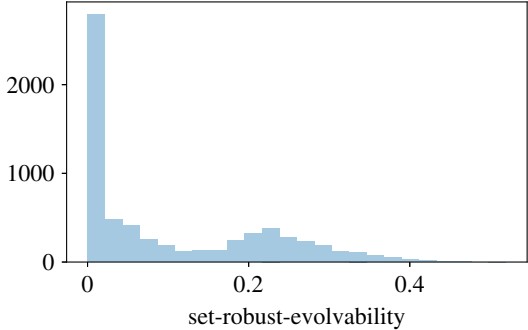

**Figure 10.** Distribution of the set-robustness and evolvability $s_p$ value in the 4 genes GP-map. The distribution is clearly bi-modal, justifying the split between low- and high-$s_p$ genotypes.

(as opposed to set-evolvability) will be severely impacted by such approximation as they do not correspond to a statistical property of the distribution.

## 5.5. Distribution of set-robustness and evolvability in the 4-genes GP-maps

(see figure 9)

## 5.6. Impact of gene duplication

(see figure 10)

# 6. Regression data

The regression have been performed using ordinary least square fit of the statsmodel package. The aim is to highlight different trend and not to claim that the observed correlation should fit a linear model. The $s_p$-threshold is set at 0.1.

| (a) Low-$s_p$ genotypes | | | | | | |
|---|---|---|---|---|---|---|
| dep. variable: | | irobustness | $R^2$: | | 0.389 | |
| model: | | OLS | Adj. $R^2$: | | 0.389 | |
| method: | | least squares | $F$-statistic: | | 2588. | |
| date: | | 28 January 2021 | prob ($F$-statistic): | | 0.00 | |
| time: | | 23:25:13 | log-likelihood: | | 5671.3 | |
| no. observations: | | 4065 | AIC: | | $-1.134 \times 10^4$ | |
| Df residuals: | | 4063 | BIC: | | $-1.133 \times 10^4$ | |
| Df model: | | 1 | | | | |
| | coef | s.e. | $t$ | $P > |t|$ | [0.025 | 0.975] |
| intercept | 0.2786 | 0.003 | 93.843 | 0.000 | 0.273 | 0.284 |
| evolvability | −0.5040 | 0.010 | −50.875 | 0.000 | −0.523 | −0.485 |
| omnibus: | | | 562.576 | Durbin-Watson: | 0.885 | |
| prob(omnibus): | | | 0.000 | Jarque-Bera (JB): | 1423.096 | |
| skew: | | | 0.776 | Prob(JB): | $9.52 \times 10^{-310}$ | |
| kurtosis: | | | 5.448 | Cond. No. | 11.4 | |

Warnings: [1] Standard errors assume that the covariance matrix of the errors is correctly specified.

## (b) High-$s_p$ genotypes

| dep. variable: | irobustness | $R^2$: | 0.244 |
|---|---|---|---|
| model: | OLS | adj. $R^2$: | 0.244 |
| method: | least squares | F-statistic: | 807.3 |
| date: | 28 January 2021 | prob (F-statistic): | $3.91 \times 10^{-154}$ |
| time: | 23:25:13 | log-likelihood: | 3048.2 |
| no. observations: | 2503 | AIC: | −6092. |
| Df residuals: | 2501 | BIC: | −6081. |
| Df model: | 1 | | |

| | coef | s.e. | $t$ | $P > |t|$ | [0.025 | 0.975] |
|---|---|---|---|---|---|---|
| intercept | 0.2281 | 0.006 | 39.192 | 0.000 | 0.217 | 0.240 |
| evolvability | 0.5064 | 0.018 | 28.412 | 0.000 | 0.471 | 0.541 |

| | | | | | | |
|---|---|---|---|---|---|---|
| omnibus: | | | 259.463 | Durbin-Watson: | 0.893 | |
| prob(pmnibus): | | | 0.000 | Jarque-Bera (JB): | 359.460 | |
| skew: | | | 0.815 | Prob(JB): | $8.80 \times 10^{-79}$ | |
| kurtosis: | | | 3.889 | Cond. No. | 13.7 | |

Data accessibility. Data and relevant code for this research work are stored in GitHub: https://github.com/vatj/gpmap_integer_polyomino.git and have been archived within the Zenodo repository: (doi:10.5281/zenodo.4533661).

Authors' contributions. V.J. and A.S.L. wrote the code and designed some of the research. V.J. performed the simulations and wrote the manuscript. S.E.A. guided the research design and the writing of the manuscript.

Competing interests. We declare we have no competing interests.

Funding. This work was supported by the Engineering and Physical Sciences Research Council (V.J. and A.S.L.) (grant nos 1642292 and 1805372), the Gatsby foundation (V.J., A.S.L. and S.E.A.) (grant no. PTAG/021). Computing facilities were provided by the Theory of Condensed Matter Group.

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
