## [Peer Review File · Royal Society Open Science]

Review History

RSOS-201636.R0 (Original submission)

Review form: Reviewer 1

Is the manuscript scientifically sound in its present form?

Yes

Are the interpretations and conclusions justified by the results?

No

Is the language acceptable?

Yes

Do you have any ethical concerns with this paper?

No

Have you any concerns about statistical analyses in this paper?

Yes

Recommendation?

Major revision is needed (please make suggestions in comments)

Comments to the Author(s)

I have now reviewed the manuscript "Gene duplication and subsequent diversification strongly affect phenotypic evolvability and robustness" by Victor Jouffrey and colleagues. In this manuscript, the authors present the results of an instance of the polyomino model, which describes in a simplified way protein-protein interactions, with two interesting additions: (i) they consider non-determinism, meaning by that that some sets of proteins can randomly form alternative structures, and (ii) they explore the consequences of gene duplication on robustness – the consistency of the phenotype after single mutations – and evolvability – the ability to generate novel shapes with single mutations.

I find myself in the difficult position of being simultaneously very enthusiastic about the study, and concerned by the doubt that some of the results are due to arbitrary choices. Also, a part of the literature on the relation between robustness and evolvability has been ignored and should be discussed. The modeling framework chosen is appropriate to investigate this question, but the extensive focus on protein-protein interaction should be discussed and perhaps questioned at some point.

I believe that a part of the results, in particular the positive relationship between robustness and evolvability, comes from the arbitrary discretization of continuous variables. Evolvability is defined as the production of a novel shape with a frequency that overcomes an arbitrary threshold, whereas robustness is the reliable production of one of the shapes produced by the ancestor. If one assumes that robustness is instead the invariance in the production of ancestral shapes, the intersection between robustness and evolvability might vanish.

While this discretisation is arbitrary, I think that a similar process takes place at another level, when the phenotype is converted into fitness. One could argue that producing the right structure, even at some low frequency, is better than not producing it; and even that some level of expression is sufficient, just like producing a non-functional form of an enzyme can sometimes be effectively neutral, thus explaining that loss-of-function mutations are often recessive. In this case, non-deterministic assembly could play an important role, by effectively decoupling robustness and evolvability. Overall I find this question really interesting, and I think the framework and the raw results are appropriate to address it. But the statistical treatment lacks the appropriate connection with biology to make a strong evolutionary argument.

My other main concern is that large parts of the literature on this question is missing. In particular and aside from the general literature on evolvability, which is only partly covered, Draghi et al (Nature, 2010) have addressed the question of the positive correlation between robustness and evolvability, using a population genetics model where population variance is considered. Rajon and Masel (Genetics, 2013) have used a quantitative genetics approach to show that evolvability increases in multilocus traits due to compensatory mutations, which I think is also relevant to discuss the link between duplication and evolvability.

Finally, I found it difficult to link the two parts of the study, namely the exploration of the relationship between robustness and evolvability and the consequences of duplications. I can see two reasons: (i) the first part is on a 4-genes model whereas the second is on 2-genes + duplication, making it difficult to connect the results. And (ii), the impact of non-determinism in part 1 is not discussed in part 2, where the examples taken seemingly involve deterministic phenotypes.

Specific comments

line 10 suppress ", more traditional"

lines 39-45 At this stage, it is difficult to understand how GP maps can be non-deterministic. Please provide an explanation.

line 8-11 I am personally not familiar with the polyomino model. Please provide a short description, perhaps focusing on why this is a good way to describe protein assembly and evolution; in particular, how do mutations in this model correspond to amino-acid changes in reality? (this is obviously essential here)

P5 L34 I think that the claim that "the restriction of a phenotype to a single Polyomino is ill-suited to the study of gene duplication " is unsubstantiated. In this example, the non-determinism in the model would implicate that the same set of subunits of globin form functionally different complexes. Is this the case? If not, such that only genetically different proteins form different higher-order structures, then the process is deterministic. I suppose that non-determinism could be selected against in some instances as this produces the expected structure with a lower probability, or that it might help evolutionary innovations by randomly producing structures that perform separate functions. And that maybe this has been involved in the evolution of the globin family. If this is what is meant here, some rewriting is required.

P5 L57 This apparent tension between robustness and evolvability can be relaxed at the population level when robustness allows genetic variance to build up and simultaneously explore different neighborhoods. See Draghi et al, Nature 2010. This should at least be discussed, if not addressed by a population genetics model...

P6 L13 Does this sentence just end here with "evolvability"?

P6 L17 I think this definition of robustness is problematic, as a change in the frequencies of polyomino produced may involve a phenotype change. Why not calculating a distance between the sets of polyomino produced, which would capture this changes in frequency? This would also avoid setting an arbitrary threshold to estimate reliability... Similarly, the calculation of genetic instability is very discrete ; why not considering instead a measure of an internal distance based on frequencies? Producing many shapes at low frequencies would be considered more unstable than producing 2, with one at very high frequency (which is arguably close to producing deterministically a single shape).

P8 L55 It is unclear what precisely is done here. Is this a comparison between single genes and pairs, before any mutation has fixed. Then why would robustness be impeded? Any single mutation should still allow the previous structure to form, since the ancestral allele remains (this corresponds to the subfunctionalization theory of Force et al). This could explain why robustness increases when s_p is small in fig.5: a deterministic set becomes more robust ; but why non-deterministic sets have decreasing robustness remains unclear to me, and is not discussed in the manuscript...

Fig. 5 Why only showing the results of 4-genes maps. Perhaps showing 1 to 3 or 4 would make sense, otherwise please justify this choice. Same comment for figures 3 (not clear if four-tile means 4 genes there) and 4. I think that the examples in figs 6-7 help understanding of how the model works, and starting the results with the 1 or 2 genes case would help a lot.

Fig. 7 I suspect a dilution effect after duplication : some of the neighboring mutations become more frequent after duplication ; if the duplicated gene contains fewer novel shapes in its

neighborhood, the overall evolvability decreases. Is that the correct explanation? If so, please consider discussing that this is due to the choice made of considering the proportion of novel phenotypes and not their absolute number, which might not change as the overall mutation rate also increases after duplication.

P10 L47 Wouldn't the loss of redundancy just bring back robustness to its pre-duplication level? The effect is much larger than this here.

Decision letter (RSOS-201636.R0)

Dear Mr Jouffrey

The Editors assigned to your paper RSOS-201636 "Gene duplication and subsequent diversification strongly affect phenotypic evolvability and robustness" have now received comments from reviewers and would like you to revise the paper in accordance with the reviewer comments and any comments from the Editors. Please note this decision does not guarantee eventual acceptance.

Please submit your revised manuscript and required files (see below) no later than 21 days from today's (ie 14-Dec-2020) date. Note: the ScholarOne system will 'lock' if submission of the revision is attempted 21 or more days after the deadline. If you do not think you will be able to meet this deadline please contact the editorial office immediately.

on behalf of Professor Andrew Simons (Associate Editor) and Pietro Cicuta (Subject Editor)
openscience@royalsociety.org

Associate Editor Comments to Author (Professor Andrew Simons):

Associate Editor: 1

Comments to the Author:

We have now received a review (in addition to the two transferred reviews) of "Gene duplication and subsequent diversification strongly affect phenotypic evolvability and robustness." Although the reviewer is positive about the approach taken and the model results themselves, substantial concerns are raised about the strength of conclusions drawn from these results. In particular, the discretization of continuous variables requires explicit justification, and the effects of this discretization on the relationship between robustness and evolvability requires full discussion.

Also needed are measures of strength and statistical confidence of the correlations presented in main result figures. Finally, as noted in this review as well as in those of the original transferred manuscript, the physical model and biology are not well connected. A fuller discussion of the existing literature on the evolutionary context of the model is needed; in particular, on the relationship between robustness and evolvability. Results should be placed clearly in context of this literature.

Reviewer comments to Author:

Reviewer: 1

Comments to the Author(s)

I have now reviewed the manuscript "Gene duplication and subsequent diversification strongly affect phenotypic evolvability and robustness" by Victor Jouffrey and colleagues. In this manuscript, the authors present the results of an instance of the polyomino model, which describes in a simplified way protein-protein interactions, with two interesting additions: (i) they consider non-determinism, meaning by that that some sets of proteins can randomly form alternative structures, and (ii) they explore the consequences of gene duplication on robustness – the consistency of the phenotype after single mutations – and evolvability – the ability to generate novel shapes with single mutations.

I find myself in the difficult position of being simultaneously very enthusiastic about the study, and concerned by the doubt that some of the results are due to arbitrary choices. Also, a part of the literature on the relation between robustness and evolvability has been ignored and should be discussed. The modeling framework chosen is appropriate to investigate this question, but the extensive focus on protein-protein interaction should be discussed and perhaps questioned at some point.

I believe that a part of the results, in particular the positive relationship between robustness and evolvability, comes from the arbitrary discretization of continuous variables. Evolvability is defined as the production of a novel shape with a frequency that overcomes an arbitrary threshold, whereas robustness is the reliable production of one of the shapes produced by the ancestor. If one assumes that robustness is instead the invariance in the production of ancestral shapes, the intersection between robustness and evolvability might vanish.

While this discretisation is arbitrary, I think that a similar process takes place at another level, when the phenotype is converted into fitness. One could argue that producing the right structure, even at some low frequency, is better than not producing it; and even that some level of expression is sufficient, just like producing a non-functional form of an enzyme can sometimes be effectively neutral, thus explaining that loss-of-function mutations are often recessive. In this case, non-deterministic assembly could play an important role, by effectively decoupling

robustness and evolvability. Overall I find this question really interesting, and I think the framework and the raw results are appropriate to address it. But the statistical treatment lacks the appropriate connection with biology to make a strong evolutionary argument.

My other main concern is that large parts of the literature on this question is missing. In particular and aside from the general literature on evolvability, which is only partly covered, Draghi et al (Nature, 2010) have addressed the question of the positive correlation between robustness and evolvability, using a population genetics model where population variance is considered. Rajon and Masel (Genetics, 2013) have used a quantitative genetics approach to show that evolvability increases in multilocus traits due to compensatory mutations, which I think is also relevant to discuss the link between duplication and evolvability.

Finally, I found it difficult to link the two parts of the study, namely the exploration of the relationship between robustness and evolvability and the consequences of duplications. I can see two reasons: (i) the first part is on a 4-genes model whereas the second is on 2-genes + duplication, making it difficult to connect the results. And (ii), the impact of non-determinism in part 1 is not discussed in part 2, where the examples taken seemingly involve deterministic phenotypes.

Specific comments

line 10 suppress ", more traditional"

lines 39-45 At this stage, it is difficult to understand how GP maps can be non-deterministic. Please provide an explanation.

line 8-11 I am personally not familiar with the polyomino model. Please provide a short description, perhaps focusing on why this is a good way to describe protein assembly and evolution; in particular, how do mutations in this model correspond to amino-acid changes in reality? (this is obviously essential here)

P5 l34 I think that the claim that "the restriction of a phenotype to a single Polyomino is ill-suited to the study of gene duplication" is unsubstantiated. In this example, the non-determinism in the model would imply that the same set of subunits of globin form functionally different complexes. Is this the case? If not, such that only genetically different proteins form different higher-order structures, then the process is deterministic. I suppose that non-determinism could be selected against in some instances as this produces the expected structure with a lower probability, or that it might help evolutionary innovations by randomly producing structures that perform separate functions. And that maybe this has been involved in the evolution of the globin family. If this is what is meant here, some rewriting is required.

P5 L57 This apparent tension between robustness and evolvability can be relaxed at the population level when robustness allows genetic variance to build up and simultaneously explore different neighborhoods. See Draghi et al, Nature 2010. This should at least be discussed, if not addressed by a population genetics model...

P6 L13 Does this sentence just end here with "evolvability"?

P6 L17 I think this definition of robustness is problematic, as a change in the frequencies of polyomino produced may involve a phenotype change. Why not calculating a distance between the sets of polyomino produced, which would capture this changes in frequency? This would also avoid setting an arbitrary threshold to estimate reliability...

Similarly, the calculation of genetic instability is very discrete ; why not considering instead a measure of an internal distance based on frequencies? Producing many shapes at low frequencies would be considered more unstable than producing 2, with one at very high frequency (which is arguably close to producing deterministically a single shape).

P8 L55 It is unclear what precisely is done here. Is this a comparison between single genes and pairs, before any mutation has fixed. Then why would robustness be impeded? Any single mutation should still allow the previous structure to form, since the ancestral allele remains (this corresponds to the subfunctionalization theory of Force et al). This could explain why robustness increases when s_p is small in fig.5: a deterministic set becomes more robust ; but why non-deterministic sets have decreasing robustness remains unclear to me, and is not discussed in the manuscript...

Fig. 5 Why only showing the results of 4-genes maps. Perhaps showing 1 to 3 or 4 would make sense, otherwise please justify this choice. Same comment for figures 3 (not clear if four-tile means 4 genes there) and 4. I think that the examples in figs 6-7 help understanding of how the model works, and starting the results with the 1 or 2 genes case would help a lot.

Fig. 7 I suspect a dilution effect after duplication : some of the neighboring mutations become more frequent after duplication ; if the duplicated gene contains fewer novel shapes in its neighborhood, the overall evolvability decreases. Is that the correct explanation? If so, please consider discussing that this is due to the choice made of considering the proportion of novel phenotypes and not their absolute number, which might not change as the overall mutation rate also increases after duplication.

P10 L47 Wouldn't the loss of redundancy just bring back robustness to its pre-duplication level? The effect is much larger than this here.

===PREPARING YOUR MANUSCRIPT===

Your revised paper should include the changes requested by the referees and Editors of your manuscript. You should provide two versions of this manuscript and both versions must be provided in an editable format:
 one version identifying all the changes that have been made (for instance, in coloured highlight, in bold text, or tracked changes);
 a 'clean' version of the new manuscript that incorporates the changes made, but does not highlight them. This version will be used for typesetting if your manuscript is accepted.
 Please ensure that any equations included in the paper are editable text and not embedded images.

If you have been asked to revise the written English in your submission as a condition of publication, you must do so, and you are expected to provide evidence that you have received language editing support. The journal would prefer that you use a professional language editing

service and provide a certificate of editing, but a signed letter from a colleague who is a native speaker of English is acceptable. Note the journal has arranged a number of discounts for authors using professional language editing services (<https://royalsociety.org/journals/authors/benefits/language-editing/>).

===PREPARING YOUR REVISION IN SCHOLARONE===

<https://royalsociety.org/journals/authors/author-guidelines/#supplementary-material> to include a suitable title and informative caption. An example of appropriate titling and captioning

may be found at https://figshare.com/articles/Table_S2_from_Is_there_a_trade-off_between_peak_performance_and_performance_breadth_across_temperatures_for_aerobic_sc_ope_in_teleost_fishes_/3843624.

Author's Response to Decision Letter for (RSOS-201636.R0)

See Appendix A.

Decision letter (RSOS-201636.R1)

Dear Mr Jouffrey

On behalf of the Editors, we are pleased to inform you that your Manuscript RSOS-201636.R1 "Gene duplication and subsequent diversification strongly affect phenotypic evolvability and robustness" has been accepted for publication in Royal Society Open Science subject to minor revision in accordance with the referees' reports. Please find the referees' comments along with any feedback from the Editors below my signature.

Please submit your revised manuscript and required files (see below) no later than 7 days from today's (ie 22-Feb-2021) date. Note: the ScholarOne system will 'lock' if submission of the revision is attempted 7 or more days after the deadline. If you do not think you will be able to meet this deadline please contact the editorial office immediately.

Kind regards,
Anita Kristiansen

Editorial Coordinator

on behalf of Professor Andrew Simons (Associate Editor) and Pietro Cicuta (Subject Editor)
 openscience@royalsociety.org

Associate Editor Comments to Author (Professor Andrew Simons):

Comments to the Author:

Revisions and responses to reviewer criticism and queries are quite complete and satisfactory.
 Only a few small issues remain:

No References section is present, and the in-text citations contain field code errors.

Two paragraphs on first page of the Introduction begin, "Over the past [three, few] decades..."

Revise.

A main criticism of Reviewer 1 was that "large parts of the literature" are missing. Although the two specific citations suggested by Reviewer 1 on correlations between robustness and evolvability have been added, citations to foundational literature on evolvability are still required.

===PREPARING YOUR MANUSCRIPT===

===PREPARING YOUR REVISION IN SCHOLARONE===

Author's Response to Decision Letter for (RSOS-201636.R1)

See Appendix B.

Decision letter (RSOS-201636.R2)

Dear Mr Jouffrey,

I am pleased to inform you that your manuscript entitled "Gene duplication and subsequent diversification strongly affect phenotypic evolvability and robustness" is now accepted for publication in Royal Society Open Science.

on behalf of Professor Andrew Simons (Associate Editor) and Pietro Cicuta (Subject Editor)
openscience@royalsociety.org

Appendix A

Associate Editor Comments to Author (Professor Andrew Simons):

Associate Editor: 1

Comments to the Author:

We have now received a review (in addition to the two transferred reviews) of “Gene duplication and subsequent diversification strongly affect phenotypic evolvability and robustness.” Although the reviewer is positive about the approach taken and the model results themselves, substantial concerns are raised about the strength of conclusions drawn from these results. In particular, the discretization of continuous variables requires explicit justification, and the effects of this discretization on the relationship between robustness and evolvability requires full discussion. Also needed are measures of strength and statistical confidence of the correlations presented in main result figures. Finally, as noted in this review as well as in those of the original transferred manuscript, the physical model and biology are not well connected. A fuller discussion of the existing literature on the evolutionary context of the model is needed; in particular, on the relationship between robustness and evolvability. Results should be placed clearly in context of this literature.

Answer:

We have amended the manuscript in the hope of providing clearer explanations in the light of the reviewer's comments. We have added measure of statistical confidence where appropriate, as well as included additional reference to underscore how our approach is complementary to existing population genetic and evolutionary dynamics models. We have also devoted efforts to underlie how the perceived discretisation issue is merely an artifact of thinking of phenotype as set of features which can be lost or preserved upon mutation.

Reviewer comments to Author:

Reviewer: 1

Comments to the Author(s)

I have now reviewed the manuscript "Gene duplication and subsequent diversification strongly affect phenotypic evolvability and robustness " by Victor Jouffrey and colleagues. In this manuscript, the authors present the results of an instance of the polyomino model, which describes in a simplified way protein-protein interactions, with two interesting additions: (i) they consider non-determinism, meaning by that that some sets of proteins can randomly form alternative structures, and (ii) they explore the consequences of gene duplication on robustness – the consistency of the phenotype after single mutations – and evolvability – the ability to generate novel shapes with single mutations.

I find myself in the difficult position of being simultaneously very enthusiastic about the study, and concerned by the doubt that some of the results are due to arbitrary choices. Also, a part of the literature on the relation between robustness and evolvability has been ignored and should be discussed. The modeling framework chosen is appropriate to investigate this question, but the extensive focus on protein-protein interaction should be discussed and perhaps questioned at some point.

Answer: We agree with the reviewer on the fact that the scope of our result is limited to protein-protein interaction due to the intrinsic limitation of the polyomino model. We added the following to emphasize why we think it is a relevant model to study gene duplication as well as explicating the limited scope of our results.

Added: "Over the past few decades, experimental and computational biologists have accumulated phylogenetic evidence of large gene families that have originated from a single original ancestor. Multiple biological processes have been identified as enabling gene duplication, complicating the

task of evolutionary biologists to model and assess its role and impact as a driver of evolution. Some gene families result in the formation of heteromeric protein complexes in which different protein subunits correspond to different members of the gene family\cite{Hardison2012, chloroFamilyComplex}. The Polyomino model is well-suited to investigate how protein subunits form complexes as well as how the duplication of one of the genes may affect the protein complex self-assembly process\cite{PereiraLeal2007}."

"While the polyomino model is limited in scope to model a protein self-assembly process, the methodology presented here has a more general scope. It is also worth noting that similar studies could be undertaken in other GP-map models. Indeed, the results presented in this paper highlight how the genetic local neighbourhoods are altered when a gene is duplicated. Other genetic mechanisms involving partial copy of a genetic sequence also exist (e.g retro-transposon). This process may also lead to alteration of the local GP-map landscape which in turns constrain the possible evolutionary dynamics."

I believe that a part of the results, in particular the positive relationship between robustness and evolvability, comes from the arbitrary discretization of continuous variables. Evolvability is defined as the production of a novel shape with a frequency that overcomes an arbitrary threshold, whereas robustness is the reliable production of one of the shapes produced by the ancestor. If one assumes that robustness is instead the invariance in the production of ancestral shapes, the intersection between robustness and evolvability might vanish.

Answer: The model and results we present focus on the relationship between robustness and evolvability at the genetic level. Most papers in the existing literature build-in a strong bias towards negative correlation by providing an exclusive outcome for the mutation : stay the same or completely change. The starting point of our model and analysis is to avoid this bias in order to understand whether this built-in anti-correlation in other models is justified. Our results show that both kind of correlation are possible.

The purpose of the logistic regression is simply to highlight two different behaviours rather than try to force a linear relationship between set-robustness and set-evolvability. Both correlations can be observed thanks to non-determinism and set-phenotype definition which are not limited to the previous exclusive outcome. So rather than building in correlation, our approach relaxes the constrain at the genetic level leading to the heterogeneity in the correlation sign and strength between the two quantities.

While this discretisation is arbitrary, I think that a similar process takes place at another level, when the phenotype is converted into fitness. One could argue that producing the right structure, even at some low frequency, is better than not producing it; and even that some level of expression is sufficient, just like producing a non-functional form of an enzyme can sometimes be effectively neutral, thus explaining that loss-of-function mutations are often recessive. In this case, non-deterministic assembly could play an important role, by effectively decoupling robustness and evolvability. Overall I find this question really interesting, and I think the framework and the raw results are appropriate to address it. But the statistical treatment lacks the appropriate connection with biology to make a strong evolutionary argument.

Answer: We have devoted special efforts to make the connection to the biology more explicit in our additions to the manuscript. In particular we added the following comments.

Added: "GP maps are a space in which evolutionary trajectories can be observed. They need to be combined with appropriate fitness functions and population dynamical models to fully describe evolutionary processes. However, the GP map does constrain the possibilities and probabilities of these trajectories. In particular, determining the robustness and evolvability of local genetic

neighbourhoods informs evolutionary models by providing insights into the probabilistic effects of evolutionary processes such as mutation or gene duplication. Indeed, one may calculate the average robustness and evolvability of all the genotypes with given phenotypes to provide the probabilities to evolve or persist upon mutation in a population dynamic model for example. The first part provides a statistical analysis of the distribution of accessible phenotypes upon a single mutation of a genotype. This step is necessary in order to characterize the local properties of the full GP map, before focusing our analysis on gene duplication. Indeed, the addition of a gene to a genotype alters its point-mutation genetic neighbourhood. It is therefore key to understand the properties prior to duplication in order to understand the consequences of this altered local neighbourhood on the accessible phenotype distribution. These results directly inform future models of evolutionary dynamics by providing insights upon how evolvability and robustness are affected by duplication events. Indeed, we show how mutation before and after duplication display different probabilities of changing the phenotype. These probabilities and their relationship are key parameters in population genetic models and evolutionary dynamic studies.

The results presented in this paper have been focused on understanding how these local properties are altered upon duplication of a gene. The metrics introduced in the methodology section provide a measure of quantities relevant to population genetic models: Set-robustness is the probability that at least one of the features in the phenotype is retained while set-evolvability measures the probability that a new feature is added to the phenotype. We have analysed how these local probabilities are altered upon duplication of a gene in our model. The results show that in average these local neighbourhoods can be affected very differently depending on the phenotype. While limited in scope to a model focused on protein assembly, they raise the question of whether models which consider duplication to have an equal impact on all individuals independently of their phenotype might be unadapted.

From a modelisation perspective, the work presented in this paper explores the consequences of phenotype defined as set in the context of GP-maps. Indeed, the framework is built around the idea that phenotype may overlap, leading to the possibility for individual mutations to both generate new features but also preserve acquired ones. Allowing for granularity at the phenotype level is bringing the modelisation one step closer to biological system as well as possibly enabling to model different evolutionary processes including those where gene duplication plays a major role."

My other main concern is that large parts of the literature on this question is missing. In particular and aside from the general literature on evolvability, which is only partly covered, Draghi et al (Nature, 2010) have addressed the question of the positive correlation between robustness and evolvability, using a population genetics model where population variance is considered. Rajon and Masel (Genetics, 2013) have used a quantitative genetics approach to show that evolvability increases in multilocus traits due to compensatory mutations, which I think is also relevant to discuss the link between duplication and evolvability.

Answer: We have added both reference to the manuscript as well as a short discussion of how the approaches are complementary and aimed at different objectives.

Added:

"This relationship between robustness and evolvability has been the focus of numerous experimental and theoretical investigations. Particular attention has been devoted to linking the local properties of GP maps at the genotype level to robustness and evolvability at the population

scale\cite{Draghi2010, Rajon1209}. While robustness and evolvability are modelled as mutually exclusive possibilities on a genotypic level, studies have shown that a positive correlation between robustness and evolvability emerges at the phenotypic scale.

The results presented here depart from previous work\cite{Draghi2010, Rajon1209} as they rely on fundamentally different assumptions. The addition of non-determinism to the model enables an overlap between robustness (conserving a phenotypic trait) and evolvability (discovering a different trait) at the genetic level. Instead of thinking of mutation or duplication as simply changing the phenotype, the focus here is on the addition or loss of phenotypic features. This also means we have to define robustness and evolvability accordingly. *

While this non-deterministic model has wider implications that would be interesting to explore, the emphasis here is on the effect of gene duplication on the structural properties of the GP map. The statistical analysis aims to provide a large-scale perspective, but does not permit conclusions at the population level. Rather, it provides a starting point for the definition of local GP map properties that work in the context of gene duplication as well as single point mutations."

* Note to the reviewer: : One can in principle follow a similar approach to [draghi] to derive the impact of such assumption at the population level. Phenotype would be define as a set of random features. .At each step one would have a probability to add one (or more) features to the phenotype (evolvability), as well as a certain probability for each of the features already present to be preserved (robustness).

Finally, I found it difficult to link the two parts of the study, namely the exploration of the relationship between robustness and evolvability and the consequences of duplications. I can see two reasons: (i) the first part in on a 4-genes model whereas the second is on 2-genes + duplication, making it difficult to connect the results. And (ii), the impact of non-determinism in part 1 is not discussed in part 2, where the examples taken seemingly involve deterministic phenotypes.

Answer: While the phenotypes are deterministic, it does not imply that the genetic neighbourhood cannot have non-deterministic genotypes. In order to make the connection to non-determinism more explicit we moved a figure from the appendix to the main text and added the following short discussion.

Added:"The results presented in figure \ref{fig:tradeoff_trimer_robustness_stability_full} suggest an interesting connection between gene duplication and non-determinism. Indeed, post-duplication data highlight an increased tendency to form rare shapes in the local neighbourhood of the genotypes that possess a duplicated gene. This is contrasted by post-specialisation data which indicates a reduced presence of genotypes forming rare shapes in the neighbourhood of specialised genotypes. The first observation is aligned with the expectation that duplication enables the accumulation of features at the phenotype level despite the stability issues, while the second observation would suggest that the specialisation step provides stability of the self-assembly process at the expense of the robustness bonus provided by redundancy. This contrast points towards an interesting and intricate link between the duplication-specialisation process and non-determinism. While only indicative, these results are in agreement with the larger studies presented in the first part of the discussion where non-determinism appeared to be distributed unevenly across the GP map."

Specific comments

line 10 suppress ", more traditional"
deleted

lines 39-45 At this stage, it is difficult to understand how GP maps can be non-deterministic. Please provide an explanation.

Added I38: In the proposed model, non-determinism refers to the possibility that some genotypes may translate into more than one Polyomino, resulting in phenotypes with a variable number of features.

line 8-11 I am personally not familiar with the polyomino model. Please provide a short description, perhaps focusing on why this is a good way to describe protein assembly and evolution; in particular, how do mutations in this model correspond to amino-acid changes in reality? (this is obviously essential here)

Added I59: This model is a coarse-grained representation of protein complex self-assembly, where proteins bind together to form larger structures. The coarse-graining procedure disregards the details of the secondary and tertiary structure of the protein as well as the molecular forces responsible for protein binding. Each three-dimensional interface is replaced by a single integer label identifying the type of interaction. Each specific label (except one neutral one) interacts with one other label. All interactions are attractive. As the internal structure of protein is lost, genetic mutation now means a mutation of the integer interaction labels, leading to the loss or gain of binding interactions between the subunits.

P5 I34 I think that the claim that "the restriction of a phenotype to a single Polyomino is ill-suited to the study of gene duplication " is unsubstantiated. In this example, the non-determinism in the model would imply that the same set of subunits of globin form functionally different complexes. Is this the case? If not, such that only genetically different proteins form different higher-order structures, then the process is deterministic. I suppose that non-determinism could be selected against in some instances as this produces the expected structure with a lower probability, or that it might help evolutionary innovations by randomly producing structures that perform separate functions. And that maybe this has been involved in the evolution of the globin family. If this is what is meant here, some rewriting is required.

Added and rephrased I119:

The deterministic version of the Polyomino model allows only one Polyomino per genotype. The heme gene family example as well as numerous other ones highlight however that we should also consider the diversification of protein complexes that can be formed, meaning the emergence of coexisting variants of a given complex through duplication and subsequent specialisation. In other words, it shows how the restriction of a phenotype to a single Polyomino is ill-suited to the study of gene duplication as the restriction to deterministic phenotypes inhibits the diversification of protein complexes

P5 L57 This apparent tension between robustness and evolvability can be relaxed at the population level when robustness allows genetic variance to build up and simultaneously explore different neighborhoods. See Draghi et al, Nature 2010. This should at least be discussed, if not addressed by a population genetics model...

Answer: Indeed this tension between robustness and evolvability is "relaxed" at the population level when taking the population robustness to be the average of the robustness of all the genotype with a given phenotype while the evolvability is redefined very differently as it is the number of unique phenotype which are in the neighbourhood of any of the genotype.

While relaxing the constrain can be justified in a population model, it does not help answer the question of the effect of duplication on the neighbourhood of a particular phenotype or address the issue at hand as the generalisation to duplication is far from obvious/straightforward. Duplication connects GP-maps with different number of accessible phenotype and different number of neighbours (as an additional gene leads to more possible mutation) leading to normalisation issues.

We are not arguing that there are no good reasons from a biological standpoint to define evolvability at a population level using a different perspective than for robustness. However, the issue at hand here is to perform a statistical analysis of how the local/accessible genetic neighbourhood is altered upon

duplication. In that sense, the population level definition of evolvability is not relevant and its single-genotype version suffers from a strong bias towards negative correlations with robustness.

P6 L13 Does this sentence just end here with "evolvability"?

Added :

The metrics introduced below aim to provide continuity with earlier work on the Polyomino model where possible, but will also diverge from this prior work due to the new definition of phenotypes as sets of Polyominoes, and also due to the focus of this study on the impact of duplication on local genetic neighbourhoods.

Removed:

We introduce the metrics set robustness and set evolvability

P6 L17 I think this definition of robustness is problematic, as a change in the frequencies of polyomino produced may involve a phenotype change. Why not calculating a distance between the sets of polyomino produced, which would capture this changes in frequency? This would also avoid setting an arbitrary threshold to estimate reliability...

Similarly, the calculation of genetic instability is very discrete ; why not considering instead a measure of an internal distance based on frequencies? Producing many shapes at low frequencies would be considered more unstable than producing 2, with one at very high frequency (which is arguably close to producing deterministically a single shape).

Answer : The reviewer raised an important point that has been a subject of some thinking prior to the publication of this paper and is a valid avenue of investigation.

Arguably part of the choice for defining set-robustness in such fashion is linked to what the question we wanted answer to : What is the probability after the mutation to have retain previous shapes?

The problem with defining the suggested distance is that discovering a new shape that you can produce reliably necessarily impact your robustness score since part of the build frequency goes to the new shape. This would introduce a similar bias towards lowering the robustness when new shapes are discovered eventhough the original shape is retained. In that sense it is arguing to build in the statistical tools analysing the GP-map the same constrain as in deterministic settings. It is arguing that adding new functionalities/shapes always come at the expense of losing some. Set-robustness answers a particular question that may not always be the best choice depending on the context of the study. Experimental work in biology points towards gene duplication and gene families to be an extremely efficient way to retain old functionalities and discover new ones.

P8 L55 It is unclear what precisely is done here. Is this a comparison between single genes and pairs, before any mutation has fixed. Then why would robustness be impeded? Any single mutation should still allow the previous structure to form, since the ancestral allele remains (this corresponds to the subfunctionalization theory of Force et al). This could explain why robustness increases when s_p is small in fig.5: a deterministic set becomes more robust ; but why non-deterministic sets have decreasing robustness remains unclear to me, and is not discussed in the manuscript...

Answer : No, e.g $\{(0,0,0,1), (0,0,0,2), (0,0,0,2)\}$: if a 2 mutates then there is indeed a fail-safe however if the one mutates it does destroy the ability to create the dimer. Additionally if a 0 becomes a 1 e.g $(0,0,1,1)$ then the trimer will always be formed.

The polyomino model is a complex mapping and as far as we have investigated the answer to why robustness decreases upon duplication-mutation is not singular. Moreover we have not succeeded to systematically identify and quantify these mechanisms. It is therefore something that will have to be investigated in future works.

Fig. 5 Why only showing the results of 4-genes maps. Perhaps showing 1 to 3 or 4 would make sense, otherwise please justify this choice. Same comment for figures 3 (not clear if four-tile means 4 genes there) and 4. I think that the examples in figs 6-7 help understanding of how the model works, and starting the results with the 1 or 2 genes case would help a lot.

Answer: Going through a full example would add considerable length to the paper. Adding more information to the paper would add further length without providing so stronger evidence as the statistics involve in lower GP-maps are performed on relatively low-numbers compare to the 4-gene map.

Fig. 7 I suspect a dilution effect after duplication : some of the neighboring mutations become more frequent after duplication ; if the duplicated gene contains fewer novel shapes in its neighborhood, the overall evolvability decreases. Is that the correct explanation? If so, please consider discussing that this is due to the choice made of considering the proportion of novel phenotypes and not their absolute number, which might not change as the overall mutation rate also increases after duplication.

Answer: Fig 10 in the appendix does provide an interesting clues as to what is happening. Indeed, it seems like while set-evolvability is lowered by duplication the proportion of unstable shape in the vicinity of the genotype with a duplicated gene is greatly increased. While this does not in anyway exclude a dilution effect lowering evolvability, it points towards something more fundamental about the stability of the building assembly in the vicinity of genotype with a duplicated gene.

We remain cautious about concluding anything more general as this adaptation of the Polyomino model lacks the necessary ingredients to investigate these questions thoroughly. However, the result presented in figure 7 as well as the rest of the paper points towards important connection between gene duplication and non-determinism which warrant further investigation in better suited models such as the binary polyomino model.

P10 L47 Wouldn't the loss of redundancy just bring back robustness to its pre-duplication level? The effect is much larger than this here.

Answer: There seems to be a confusion here about what is meant by a loss of redundancy. The intent here is to highlight that while in the genotype with a duplicate gene both interface are interchangeable. Meaning if the 1 label of one of the two gene is swapped it does not prevent formation of the trimer. In the specialised case multiple effects are at play: one is loss of redundancy meaning the interface are not equivalent anymore and the duplicate genes are also not equivalent. Meaning mutation of either the 3 or the 1 leads to no trimer. Additionally there is an interference effect which is due to mutation creating an opening for the third tile to bind to other site for example (see 0041 0003 0024 forms a tetramer where as 0041 0022 would have been a trimer). This is also due to less mutation being effectively neutral (indeed 4 in the original genome cannot interact with any of the already present label while it can in the specialised version). Amended the manuscript to clarify this point.

Added : One must also consider a potential "interference" effect due to the third tile introduced by the duplication-specialisation chain. Indeed mutations previously contributing to robustness are now leading to a different shape due to the presence of the third tile. As an example consider mutating a neutral 0 label to a 4 in the original genotype versus the specialised version. The presence of the third tile leads to a tetramer in the specialised version whereas it does not impact the original one.}

Appendix B

Hello,

We have fixed the minor change in the introduction. We are also planning to make sure the reference section shows up when compiling on the rsos platform this time around ensuring as well as ensuring they are no missing reference this time around.

We have exercised our best judgement to add to the bibliography articles which pioneered studies on balancing robustness and evolvability as well as adding these references to the main text.

Robustness and evolvability: a paradox resolved, A. Wagner, 2008, Proceedings of the Royal Society B

Robustness And Evolvability in Living Systems, A. Wagner, 2013, Princeton University Press

The effect of genetic robustness on evolvability in digital organisms, E. Santiago and R. Sanjuán, 2008, BMC Evolutionary Biology

Balancing Robustness and Evolvability, R. Lenski and J. E. Barrick and C. Ofria, 2006, PLOS Biology

Thank you for your time,
V. Jouffrey